# Overcoming the permeability-selectivity challenge in water purification using two-dimensional cobalt-functionalized vermiculite membrane

Mengtao Tian[1], Yi Liu[1,2], Shaoze Zhang[3], Can Yu[4], Kostya (Ken) Ostrikov[5] & Zhenghua Zhang [1,2,5] ✉

Clean water and sanitation are major global challenges highlighted by the UN Sustainable Development Goals. Water treatment using energy-efficient membrane technologies is one of the most promising solutions. Despite decades of research, the membrane permeability-selectivity trade-off remains the major challenge for synthetic membranes. To overcome this challenge, here we develop a two-dimensional cobalt-functionalized vermiculite membrane (Co@VMT), which innovatively combines the properties of membrane filtration and nanoconfinement catalysis. The Co@VMT membrane demonstrates a high water permeance of 122.4 $L \cdot m^{-2} \cdot h^{-1} \cdot bar^{-1}$, which is two orders of magnitude higher than that of the VMT membrane (1.1 $L \cdot m^{-2} \cdot h^{-1} \cdot bar^{-1}$). Moreover, the Co@VMT membrane is applied as a nanofluidic advanced oxidation process platform to activate peroxymonosulfate (PMS) for degradation of several organic pollutants (dyes, pharmaceuticals, and phenols) and shows excellent degradation performance (~100%) and stability (for over 107 h) even in real-world water matrices. Importantly, safe and non-toxic effluent water quality is ensured by the Co@VMT membrane/PMS system without brine, which is totally different from the molecular sieving-based VMT membrane with the concentrated pollutants remaining in the brine. This work can serve as a generic design blueprint for the development of diverse nanofluidic catalytic membranes to overcome the persistent membrane permeability-selectivity issue in water purification.

Fresh water scarcity is a global planetary challenge, which might escalate in the near future due to uncontrolled population growth, climate change, and water pollution[1,2]. The urgent demand for increasing water supply beyond the hydrological cycle has necessitated advanced treatment processes aiming to raise the supply of fresh water from unconventional sources such as seawater and various types of wastewater. Compared with the traditional distillation, evaporation, adsorption, and degradation methods, state-of-the-art membrane technologies have attracted strong interest in advanced water treatment processes given their low energy consumption, high efficacy, manufacturing scalability, and small land-use and carbon emissions footprints[3]. However, the membrane permeability-selectivity trade-off is still the major hurdle for commercial applications of synthetic membranes, wherein permeability limits the flow rate while selectivity affects the products of the separation process[4].

Growing availability of nanomaterials and development of membrane fabrication technologies have stimulated intense research to overcome the membrane permeability-selectivity trade-off. Two-dimensional (2D) materials have unique physicochemical properties, atomic thickness, large aspect ratio, and chemical flexibility. This is why many 2D materials such as graphene, graphene oxide (GO), MXene, molybdenum disulfide ($MoS_2$), and boron nitride (BN), have been recently employed for the fabrication of 2D materials-based membranes to overcome the permeability-selectivity trade-off[5]. There are two kinds of 2D materials-based membranes: (i) laminar/lamellar membranes with interlayer/intralayer galleries synthesized by stacking 2D nanosheets, and (ii) nanoporous membranes prepared using intrinsic crystalline porous 2D nanosheets/drilling nano-pores on atomically thick 2D nanosheets[6–11]. 2D porous membranes can achieve precise mass separation by judicious adjustment of nanoscale pores and their distribution on monolayer 2D nanosheets through either reactive ion etching on 2D nanosheets or optimizing the crystalline porosity of 2D materials such as covalent organic frameworks (COF) and metal organic frameworks (MOF). Moreover, the atomic thickness of 2D materials reduces the resistance for mass transport while maximizing the permeation flux of 2D porous membranes[12,13]. However, the difficulty in construction of uniform and well-dispersed nanopores and the high economic cost of drilling restrict the development of 2D porous membranes[6,9]. In addition, 2D organic framework materials (e.g., COF, MOF) lack robust mechanical strength to form large surface area membranes. These thin-film materials also struggle with the existing intrinsic or extrinsic defects (e.g., grain boundaries) between single crystalline units, leading to non-selective transport[12,13].

On the other hand, 2D laminar membranes offer the possibility of flexible control over permeant passage (particularly interlayer channels) to balance the selectivity and permeability requirements. Intercalation of guest species to stabilize or broaden the interlayer spacing is one of the most promising methods for 2D laminar membranes to overcome the permeability-selectivity trade-off[14,15]. However, the loose microstructure of 2D laminar membranes compromises the frame stability by promoting the delamination of nanosheets or the compaction of laminates under typical operating conditions. Water flux through narrow nanochannels in 2D laminar membranes is low and requires extra spacer species to improve[15]. The irregular stacking of 2D nanosheets with microporous defects also limits the water-solute selectivity of 2D laminar membranes[14–16]. Furthermore, only a few reliable techniques for exfoliating high aspect ratio and intact 2D monolayers from bulk crystals are currently available, while the existing synthesis methods are not environment-friendly and involve strong acids or bases, cumbersome oxidation, and reduction processes[9,10]. Collectively, these factors pose major challenges for the development and applications of 2D laminar membranes.

To address these challenges, here we report an innovative approach to overcome the membrane permeability-selectivity trade-off using a nanoconfinement catalytic process enabled by the judiciously designed 2D cobalt-functionalized vermiculite membrane. Cobalt was chosen as the most effective activator for peroxymonosulfate to produce reactive oxygen species (ROS). The proposed membrane-based nanoconfined heterogeneous catalysis approach relies on our proprietary multi-functional membrane that can sustain both the advanced oxidation processes (AOPs) and membrane filtration. The catalysts loaded in the interlayer and/or intralayer of the 2D laminar membrane improved the water flux by increasing the spacing of the interlayer and/or intralayer, while simultaneously ensuring the effective degradation and mineralization of organic pollutants[17,18]. This approach is generic and (i) offers insights to produce multi-functional membranes to overcome the membrane permeability-selectivity trade-off; (ii) introduces new insights into the mechanisms of nanofluid mass transfer and catalysis within membrane angstrom/nanometer-confined spaces, and (iii) can be used for the processing of a broader range of water pollutants with long-term stable operation and safe, non-toxic effluent water.

## Results and discussion
### Synthesis and microanalysis of Co@VMT membrane
Co@VMT nanosheets were synthesized by a two-step process: (1) monolayer VMT synthesis, and (2) cobalt-functionalization. Bulk VMT was composed of adjacent nanosheets stacked into bulk macro-structures through electrostatic attraction with interlayer cations (Supplementary Fig. 1), which was exfoliated into monolayer VMT nanosheets via ion exchange together with hydrogen peroxide intercalation. The thickness and average lateral size of the 2D VMT nanosheets measured by atomic force microscopy (AFM) were 1.02 nm and 300 nm, respectively, confirming the monolayer structure of the 2D VMT nanosheets (Fig. 1a). The flexible and smooth monolayer morphology of the VMT nanosheets was revealed by transmission electron microscopy (TEM) (Fig. 1b). VMT flakes with negative charge of −44.9 mV originating from the deprotonation of oxygen functional groups could easily adsorb Co ions with positive charge for hydrolysis and nucleation of Co nanoparticles on the VMT surface, which was confirmed by the changed zeta potential of −17.9 mV for Co@VMT nanosheets (Supplementary Fig. 2a)[19]. This strong electrostatic interaction enabled the resulting small Co nanoparticles to be pinned on oxygen functional groups of VMT surface, effectively preventing agglomeration of Co nanoparticles[20]. Previous studies have shown that heterogeneous nucleation of Co nanoparticles on oxygen functional groups is a spontaneous and thermodynamically favorable process[15]. Co nanoparticles with the uniform size of 2–4 nm were homogeneously dispersed on the VMT surface (Fig. 1c, d). The thickness of Co@VMT nanosheets was 1.04 nm (Supplementary Fig. 2b), which was almost the same to that of VMT nanosheets (1.02 nm) (Fig. 1a). Finally, the functionalized Co@VMT hybrid nanosheets were assembled into the Co@VMT membrane by vacuum filtration, and then dehydroxylated at 130 °C (Supplementary Fig. 3).

We then analyzed the physicochemical characteristics of Co@VMT membrane. As shown in the X-ray diffraction (XRD) pattern of VMT membrane in Fig. 1e, a single high-intensity peak for $d_{(002)}$ plane appeared at 2θ of 6.61° (The interlayer free spacing is 3.04 Å, which was calculated as the difference between $d$ spacing (13.24 Å) and monolayer VMT nanosheet thickness (1.02 nm) measured by AFM in Fig. 1a), which indicated the successful conversion of the bulk phase to monolayer VMT nanosheets[19]. This characteristic diffraction peak of Co@VMT membrane shifted to a smaller 2θ of 5.89° ($d$ = 14.9 Å, 4.76 Å of the interlayer free spacing), demonstrating that the rigid Co nanoparticles expanded the interlayer spacing of the 2D laminar VMT membrane[14,15]. The XRD pattern of Co@VMT membrane showed two other typical characteristic diffraction peaks at 13.5° and 19.04°, which were attributed to $d_{(003)}$ lattice plane of α-$Co(OH)_2$ and $d_{(111)}$ lattice plane of $Co_3O_4$, respectively[21]. Other diffraction peaks of Co@VMT membrane were in a good agreement with the crystallographic structure of $Co_3O_4$ (JCPDS: #97-002-7497; Supplementary Fig. 4). The lattice fringes corresponding to the (100) and (110) planes of α-$Co(OH)_2$ and the (400) plane of $Co_3O_4$ on Co@VMT membrane suggested that Co nanoparticles were crystalline, as seen from the high-resolution transmission electron microscopy (HRTEM) images (Fig. 1f)[18].

In addition, Co@VMT membrane exhibited a relatively smooth surface with minimal wrinkling and was largely free from any pinholes (Fig. 1g). Moreover, the membrane's cross-section displayed a typical laminar structure similar to other 2D material membranes (Fig. 1h). Energy dispersive spectroscopy (EDS) mapping showed a uniform distribution of the primary elements including Co (12.43 wt%), Si, Fe, and Al through the whole membrane cross section (Supplementary Table 1). Furthermore, X-ray photoelectron spectroscopy (XPS) also confirmed the presence of Co oxides in the Co@VMT membrane

(Fig. 1i)[17,22,23]. The Co 2p spectrum exhibited characteristic peaks of Co $2p_{3/2}$ and Co $2p_{1/2}$ with the pairs of mixed oxidation states of Co$^{3+}$ (795.95 eV and 780.18 eV) (37.69%) and Co$^{2+}$ (782.18 eV and 797.48 eV) (62.31%) (Supplementary Table 2).

Further evidence to support the XPS results regarding the presence of both Co(II) and Co(III) oxides in Co@VMT membrane was provided by synchrotron X-ray absorption spectroscopy (XAS). As shown in Fig. 1j, the Co K-edge X-ray absorption near-edge structure (XANES) spectrum of Co@VMT membrane was located between that of CoO and Co$_3$O$_4$ references and was drastically different from that of the Co foil. This result indicated that Co atoms were not metal clusters but carried positive charges and their oxidation states ranged between +2 and +3[22]. The extended X-ray absorption fine structure spectroscopy (EXAFS) of Co@VMT membrane showed two peaks at 1.41 Å and 2.37 Å, which corresponded to the Co−O and Co−Co bonds, respectively (Fig. 1k). Curve fitting of the Co K-edge EXAFS indicated that the average coordination numbers of Co−O and Co−O−Co were 5.74 and 0.68, respectively (Supplementary Fig. 5 and Supplementary Table 3).

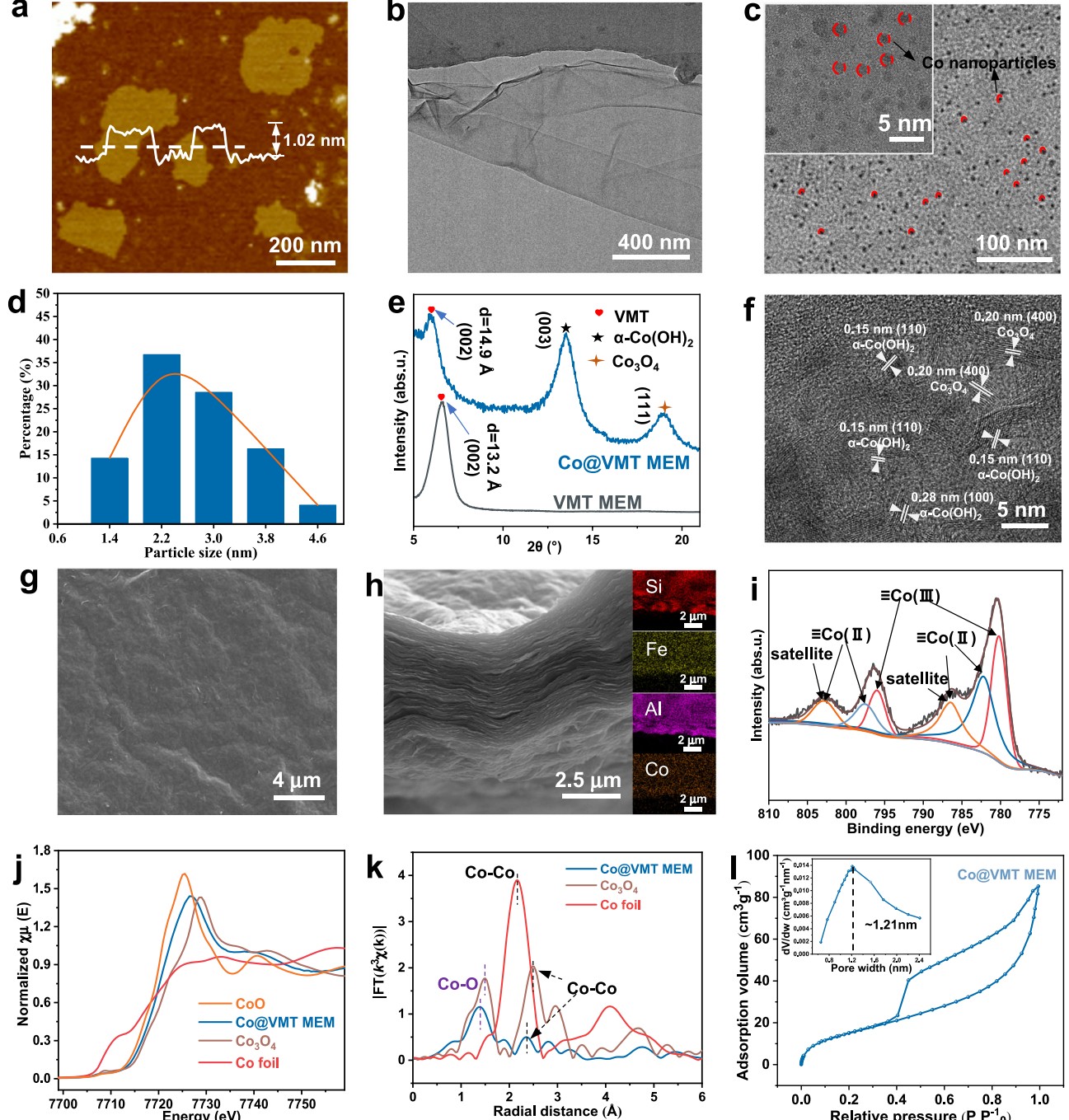

**Fig. 1 | Synthesis and microanalysis of Co@VMT membrane. a** AFM image with the height profile, **b** TEM image, **c** HRTEM image and enlarged HRTEM (inset) image of Co@VMT nanosheets. **d** Average size of Co nanoparticles on Co@VMT nanosheets (Data were obtained from the statistics of Co nanoparticles within the enlarged HRTEM image of Co@VMT nanosheets); **e** XRD patterns of wetted VMT and Co@VMT membranes at low Bragg angles; **f** HRTEM image of freestanding Co@VMT membrane; **g** SEM image of surface and **h** cross-section and EDS mapping of Co@VMT membrane; **i** Co 2p XPS spectra of Co@VMT membrane. **j** Normalized XANES spectra of Co@VMT membrane, Co foil, CoO, and Co$_3$O$_4$ references at the Co K-edge; **k** FT-EXAFS spectra of Co@VMT membrane (without phase correction). **l** Adsorption and desorption curves of Ar as well as pore size distribution (inset) for Co@VMT membrane.

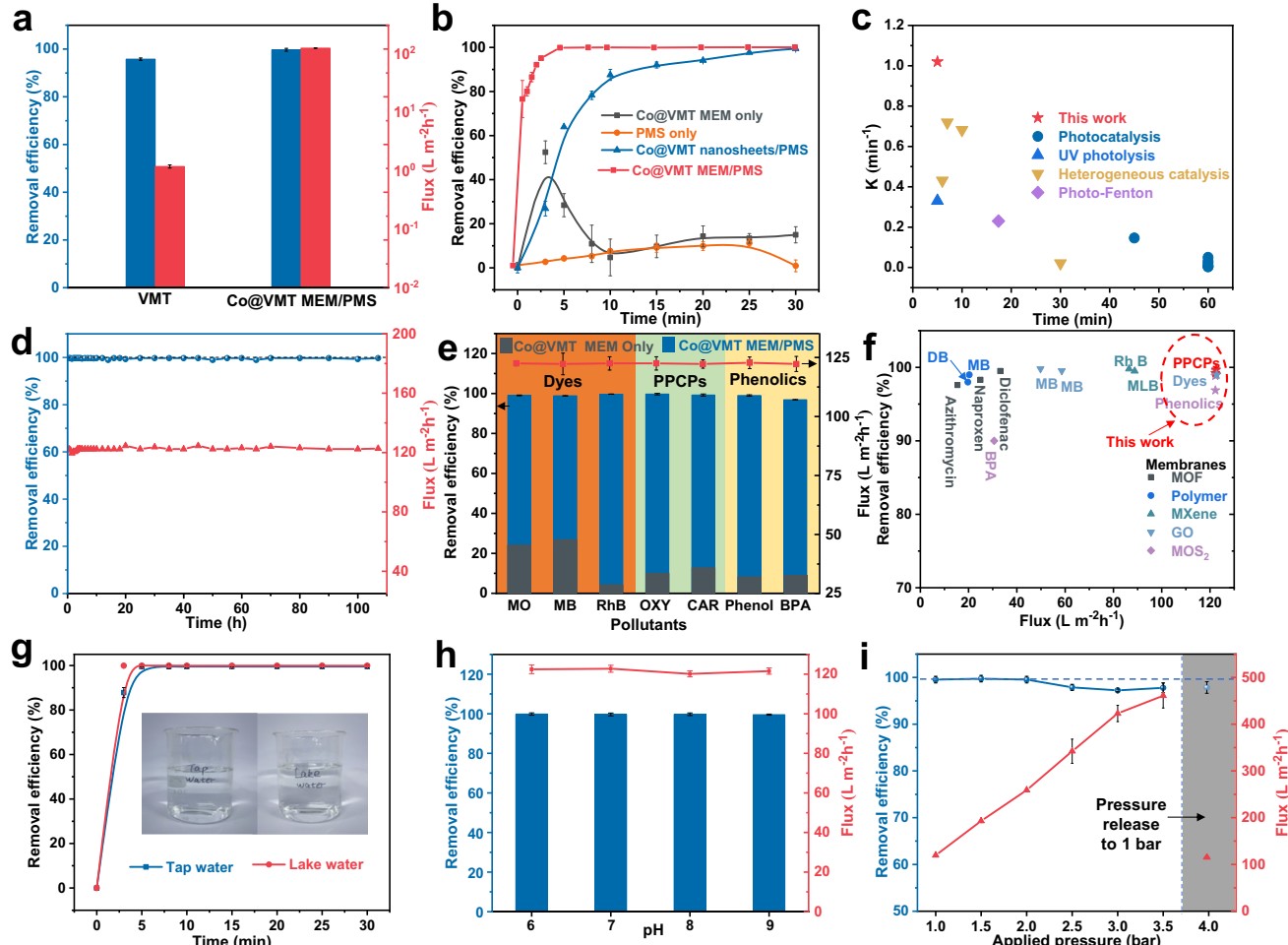

**Fig. 2 | Membrane permeability-selectivity performance evaluation of the Co@VMT membrane/PMS system. a** Flux and ranitidine removal efficiency of VMT membrane and Co@VMT membrane/PMS systems. **b** Ranitidine removal efficiency in different catalytic systems. **c** Comparison of the first-order rate constants ($k$) of different systems. Details in Supplementary Table 5. **d** Stability test of flux and removal efficiency with operation duration. **e** Removal of different organic pollutants by Co@VMT membrane/PMS system. **f** Comparison of flux and pollutant removal efficiency between Co@VMT membrane/PMS system and previously developed membranes. Details in Supplementary Table 7. Effects of **g** real water, **h** solution pH, and **i** applied pressure on ranitidine removal by Co@VMT membrane/PMS system. (Conditions of the feed solution for all membrane tests: [Ranitidine] or [Other pollutants] = 10 ppm, [PMS] = 20 ppm, pH = 4.0, Pressure = 1 bar, $T$ = 298 K). Error bars represent standard deviation of three measurements on the same membrane.

This finding was different from the coordination environment of pure $Co_3O_4$ or $Co(OH)_2$ modules, indicating the effect of interaction of Co atoms with oxygenous groups attached to VMT nanosheets[24].

To investigate the specific surface area ($S_{BET}$) and pore size distribution (PSD), the Ar adsorption and desorption curves of Co@VMT membrane were analyzed by Brunauer-Emmett-Teller (BET) and Barrett-Joyner-Halenda (BJH) methods. As shown in Fig. 1l, the adsorption curve revealed the presence of micropores within the Co@VMT membrane and the desorption isotherm was the typical H4 type for materials containing narrowly fractured pores[25,26]. Compared with the VMT membrane, the $S_{BET}$ of Co@VMT membrane increased from 5.53 to 48.27 $m^2 \cdot g^{-1}$ (Supplementary Table 4). The total pore volume also increased from 0.015 (VMT membrane) to 0.108 $cm^3 \cdot g^{-1}$ (Co@VMT membrane), respectively. High values of $S_{BET}$ are essential for the exposure of catalytic sites on surface and edges of Co@VMT membrane to improve the activation efficiency of PMS, while the increase in the total pore volume can promote the flux and ensure sufficient reaction between pollutants and ROS. The PSD of Co@VMT membrane ranged from 0.65 to 2.4 nm with the peak pore size of 1.21 nm (Fig. 1l inset), which is higher than the peak pore size (1.19 nm) of VMT membrane (Supplementary Fig. 6). This result implied that the heterogeneous nucleation of Co nanoparticles expanded both the interlayer spacing (Fig. 1e) (3.04 to 4.76 Å) and the intralayer spacing (Fig. 1l) (1.19 to 1.21 nm) of the 2D laminar VMT membrane[15,20].

**Overcoming the membrane permeability-selectivity trade-off**
Pure VMT and Co@VMT membranes with different mass ratios (Co:VMT = 1:10, 2:10, 4:10) were labeled as VMT MEM, Co@VMT-1 MEM, Co@VMT MEM, and Co@VMT-4 MEM. Their permeability-selectivity performances were explored by a probe pollutant (ranitidine) in a dead-end filtration device[17]. A typical membrane permeability-selectivity trade-off effect was observed. As shown in Fig. 2a, VMT MEM achieved a high removal efficiency of ranitidine (95.8%) and a low water permeance (1.1 $L \cdot m^{-2} \cdot h^{-1} \cdot bar^{-1}$). In contrast, loading of different amounts of Co nanocatalysts on pure VMT membranes significantly increased the membrane flux. Specifically, Co@VMT MEM with Co mass loading of 0.16 $mg \cdot cm^{-2}$ acquired a high water permeance of 122.4 $L \cdot m^{-2} \cdot h^{-1} \cdot bar^{-1}$ (two orders of magnitude higher than that of the VMT MEM). However, the rejection rate of Co@VMT MEM significantly decreased to only 14.96% (Supplementary Fig. 7). This ubiquitous trade-off effect manifested by the increase in water flux and decrease in selectivity for Co@VMT MEM conformed to the size sieving effect[15,16].

Coupling Co@VMT MEM with PMS can enable effective water filtration ensured by the unique membrane nanochannel-confined catalysis leading to high water flux and complete degradation and mineralization of organic contaminants at the same time[17,18,22,23]. Indeed, the Co@VMT MEM/PMS system achieved a removal efficiency of 100% and a high water permeance of 122.4 L·m$^{-2}$·h$^{-1}$·bar$^{-1}$ (Fig. 2a). Since membrane size exclusion and adsorption can also contribute to the removal of contaminants in the membrane system, additional experiments were carried out to investigate the related contribution to pollutant removal by the Co@VMT MEM/PMS system (Fig. 2b). For Co@VMT MEM alone, the removal of ranitidine was considerable within 5 min due to adsorption. However, the removal efficiency decreased to 12.3% after 30 min due to adsorption saturation, indicating that both adsorption and size exclusion of Co@VMT MEM were not the main contributors for the removal of ranitidine[23]. On the other hand, degradation using only PMS demonstrated low reactivity to ranitidine degradation, with a removal efficiency of only 18.5% after 30 min. The removal efficiency of about 100% for the Co@VMT nanosheets/PMS heterogeneous catalysis system after 30 min confirmed that oxidative degradation played the key role in ranitidine degradation. Noteworthy, the time taken for the complete degradation of ranitidine by the Co@VMT MEM/PMS system (5 min) was much shorter than 30 min for the Co@VMT nanosheets/PMS heterogeneous catalysis system. Moreover, the fitted first-order rate constant ($k$) of the Co@VMT MEM/PMS system was calculated to be 1.02 min$^{-1}$, which was 6.25 times faster compared to the Co@VMT nanosheets/PMS heterogeneous catalysis system (0.163 min$^{-1}$) (Supplementary Fig. 8). The first-order rate constant herein was also much better than the previously reported values for other catalytic systems (Fig. 2c and Supplementary Table 5). These experimental results demonstrated that Co@VMT MEM coupled with PMS could indeed overcome the membrane permeability-selectivity trade-off with a high removal efficiency of 100% and a high water permeance of 122.4 L·m$^{-2}$·h$^{-1}$·bar$^{-1}$. Moreover, assembling nanosheets into membrane forms could address the issues like catalyst agglomeration, recycling, and reuse during heterogeneous catalytic reactions in batch suspension solutions. As such, the Co@VMT MEM/PMS system would be more cost-effective and was preferred compared to the Co@VMT nanosheets/PMS system.

The mineralization rate is the key indicator of whether the organic pollutants can be completely transformed to water and carbon dioxide in the oxidative degradation process. The performance of ranitidine mineralization was poor in the Co@VMT nanosheets/PMS system (11.49%) and PMS only system (5.84%) (Supplementary Fig. 9). In contrast, the Co@VMT MEM/PMS system achieved a high mineralization rate of 70.9% (Supplementary Fig. 9), indicating that it can effectively reduce the amount of degradation by-products and thus achieve more efficient water purification. In addition, the PMS decomposition efficiency for the Co@VMT MEM/PMS system (75.6%) was also significantly higher compared to the other two systems (37.54% for Co@VMT nanosheets/PMS system and 11.23% for PMS only system) (Supplementary Fig. 10). This result demonstrated the high catalytic efficiency of the Co@VMT MEM/PMS system to activate PMS for ROS production[23]. Moreover, the interlayers (4.76 Å) and intralayers (1.21 nm) of Co@VMT MEM could offer abundant AOP confinement spaces, wherein the migration distances between ROS and pollutants were greatly shorten and the collisions and reactions between ROS and pollutants were significantly strengthened. As such, a significantly high ranitidine mineralization rate was achieved by the Co@VMT MEM/PMS system with the more detailed mechanisms elucidated in the following part.

Robust operation for up to 107 h with a stable water permeance of 122.4 L·m$^{-2}$·h$^{-1}$·bar$^{-1}$ and a near 100% ranitidine degradation efficiency was confirmed by the pressure-driven continuous flow experiment (Fig. 2d). The stable water flux for the Co@VMT MEM/PMS system can

be ascribed to oxidative degradation, which effectively reduced the concentration polarization and membrane fouling effects compared with the traditional membranes operated using the size exclusion mechanism[18]. At the same time, firm bonding of Co on VMT nanosheets ensured that the concentration of Co (1.6 µg.L$^{-1}$) leaked in the permeate solution was negligible (Supplementary Table 6).

The universal catalytic performance of the Co@VMT MEM/PMS system was further investigated in the removal of different organic pollutants (Fig. 2e). Remarkably, the Co@VMT MEM/PMS nanoconfinement catalysis system achieved 100% removal rates of organic pollutants with different molecular weights and functional groups such as methyl orange (MO), methylene blue (MB), rhodamine B (RhB), phenol, bisphenol A (BPA), carbamazepine (CAR), and oxytetracycline (OXY). This result is a major step change compared with the single Co@VMT MEM filtration system which showed only 4.3−27.1% removal rates. The achieved effective pollutant degradation was due to ROS produced by the Co@VMT MEM/PMS system, which is a more flexible and effective approach than adjusting the membrane pore size to achieve the removal of pollutants[17,18,22,23]. As shown in Fig. 2f, in contrast to some cutting-edge membrane systems and membrane-based AOPs systems, our proprietary system showed significant advantages in both water flux and removal efficiency (Supplementary Table 7).

Tests under diverse and harsh operating conditions were further carried to demonstrate the feasibility of the developed Co@VMT MEM/PMS system for practical applications. First, the potential application for real water was evaluated by using tap water and lake water containing some anions and natural organic matter (Supplementary Table 8), which usually inhibit the degradation efficiency. Surprisingly, the Co@VMT MEM/PMS system still showed about 100% removal efficiency for ranitidine in these two different water matrices (Fig. 2g). However, the negative effect for real water samples was reflected in the pollutant mineralization rate (Supplementary Fig. 11). The ranitidine mineralization rate decreased from 70.9% (Milli-Q water) to 68.4% (tap water) and 44.4% (lake water), which was mainly due to the competitive consumption of ROS by anions and natural organic matter, especially in lake water (Supplementary Table 8)[23]. Next, the impact of different pH conditions on the performance of the Co@VMT MEM/PMS system was investigated (Fig. 2h). When the solution pH was raised from 6 to 9, the degradation efficiency of ranitidine remained at 100% and a steady water permeance of 122.4 L·m$^{-2}$·h$^{-1}$·bar$^{-1}$ was maintained, demonstrating the remarkable adaptability of the Co@VMT MEM/PMS system toward pH variation without precipitation occurred.

In addition, the water flux increased almost linearly with the operating pressure and still maintained a relatively stable and high removal efficiency (Fig. 2i). This result suggested that the membrane structure did not collapse under continuously increasing pressure while avoiding holes and defects. On the other hand, increasing the membrane water flux shortened the retention time of pollutants in the membrane, potentially diminishing the removal efficiency of pollutants. The removal efficiency of ranitidine remained stable at ~100% when the water flux increased from 122.4 to 258.9 L·m$^{-2}$·h$^{-1}$ (Fig. 2i). However, the further increase of water flux resulted in the decrease of removal efficiency. When the water flux increased to 342.1 L·m$^{-2}$·h$^{-1}$, the removal efficiency of ranitidine decreased to 97.8% (Fig. 2i). As such, the membrane water flux should be less than 258.9 L·m$^{-2}$·h$^{-1}$ in order to achieve a 100% removal efficiency of ranitidine. When the pressure recovered down to 1 bar, the membrane exhibited almost the same flux and removal efficiency as before, thus indicating the regular layer structure and excellent mechanical properties of Co@VMT MEM[27].

## Molecular mechanisms for resolving the permeability-selectivity trade-off

To confirm the effective reactions between ROS and target pollutants as well as the fast water transport, we used molecular dynamics (MD)

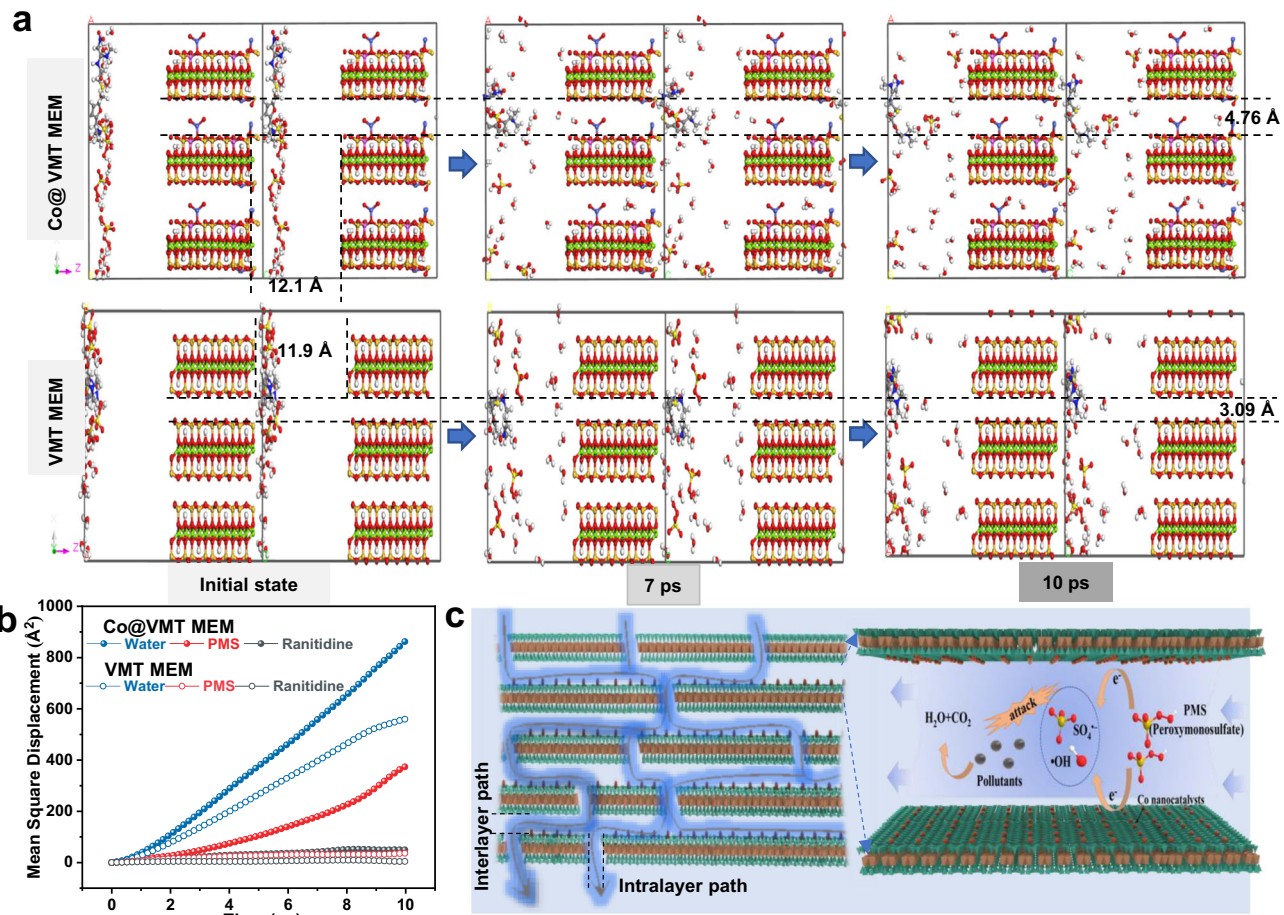

**Fig. 3 | MD simulations of mass transfer within VMT and Co@VMT membranes.** **a** MD simulations of the diffusion of PMS, ranitidine, and $H_2O$ molecules within VMT and Co@VMT membranes. **b** MSD curves of PMS, ranitidine, and $H_2O$ molecules within VMT and Co@VMT membranes. **c** Schematic diagram of mass transfer within Co@VMT membrane and Co@VMT MEM/PMS system for degradation of organic pollutants.

simulations to elucidate the diffusion of ranitidine, water, and PMS within the Co@VMT membrane nanochannels[17,28]. Ranitidine, water, and PMS can follow both the intralayer and interlayer diffusion paths within the membrane nanochannels. As shown in Fig. 3a and Supplementary Movie 1, the narrow interlayer free spacing (3.09 Å) of VMT MEM could barely allow the mass transfer of water, ranitidine ($0.570 \times 0.460 \times 1.68$ nm), and PMS ($0.315 \times 0.305 \times 0.350$ nm) (Supplementary Fig. 12) within interlayer paths, which resulted in the increased mass transfer resistance and a low flux (Fig. 2a) with intralayers as the main mass transfer paths[15,27]. However, the enlarged interlayer free spacing (4.76 Å) of the Co@VMT MEM not only enabled the transport of water and PMS molecules through the interlayer paths, but also boosted the mass transfer through the downward intralayer paths with a free spacing of 1.21 nm working as the predominant mass transport channel for large ranitidine molecules (Supplementary Movie 2). Therefore, a high flux was achieved, two orders of magnitude higher than that of the VMT MEM (Fig. 2a). The Mean Square Displacement (MSD) curves (Fig. 3b) displayed a significant increase in the mass transfer rate through Co@VMT MEM compared with VMT MEM, especially for water and PMS molecules. Water exhibited the fastest mass transfer in nanochannels, which may be attributed to high capillary pressures and the highly ordered structure of water in nano-spaces[17,29,30]. The above MD results indicate that PMS could swiftly interact with the Co catalytic sites in both the interlayer and intralayer paths of Co@VMT MEM (Fig. 3a), contributing to the efficiently catalytic decomposition of PMS molecules into ROS as well as collisions and reactions between ROS and pollutants (Fig. 3c).

Moreover, the confinement spaces provided by the interlayer/intralayer-confined nanochannels within the Co@VMT MEM can significantly reduce the migration distances between ROS and pollutants and thus remarkably facilitate the utilization of ROS (Fig. 3c)[17], resulting in the achieved efficient degradation and mineralization of organic pollutants (Fig. 2).

We emphasize that the Co@VMT MEM provided abundant and flexible nanochannels-confined spaces where electronic interactions, enrichment of ROS and catalytic reactions were triggered to achieve the efficient removal of pollutants[28]. The catalytic mechanism of the Co@VMT MEM/PMS system was revealed through density functional theory (DFT) calculations and a series of dedicated experiments including electron paramagnetic resonance (EPR) and ROS scavenging experiments.

The DFT calculations were performed to evaluate the PMS activation performance by α-Co(OH)$_2$ and Co$_3$O$_4$ phases within the Co@VMT MEM. All the calculations regarding the Co@VMT MEM nanoconfinement effect on PMS activation were performed under the intralayer free spacing of 1.21 nm, which was the predominant mass transport channel for large ranitidine molecules and was the main site for AOP reactions according to MD simulations (Fig. 3a and Supplementary Movie 2). The adsorption energy ($E_{ads}$) values of PMS molecules on the (100) and (110) planes of α-Co(OH)$_2$ and (111) plane of Co$_3$O$_4$ were −3.93, −4.48 eV and −3.32 eV, respectively, implying spontaneous activation processes of PMS by α-Co(OH)$_2$ and Co$_3$O$_4$ (Fig. 4a). Following the adsorption, the O−O bond length ($d_{O\text{-}O}$) in the adsorbed PMS molecule extended from 1.356 (PMS alone) to 1.466

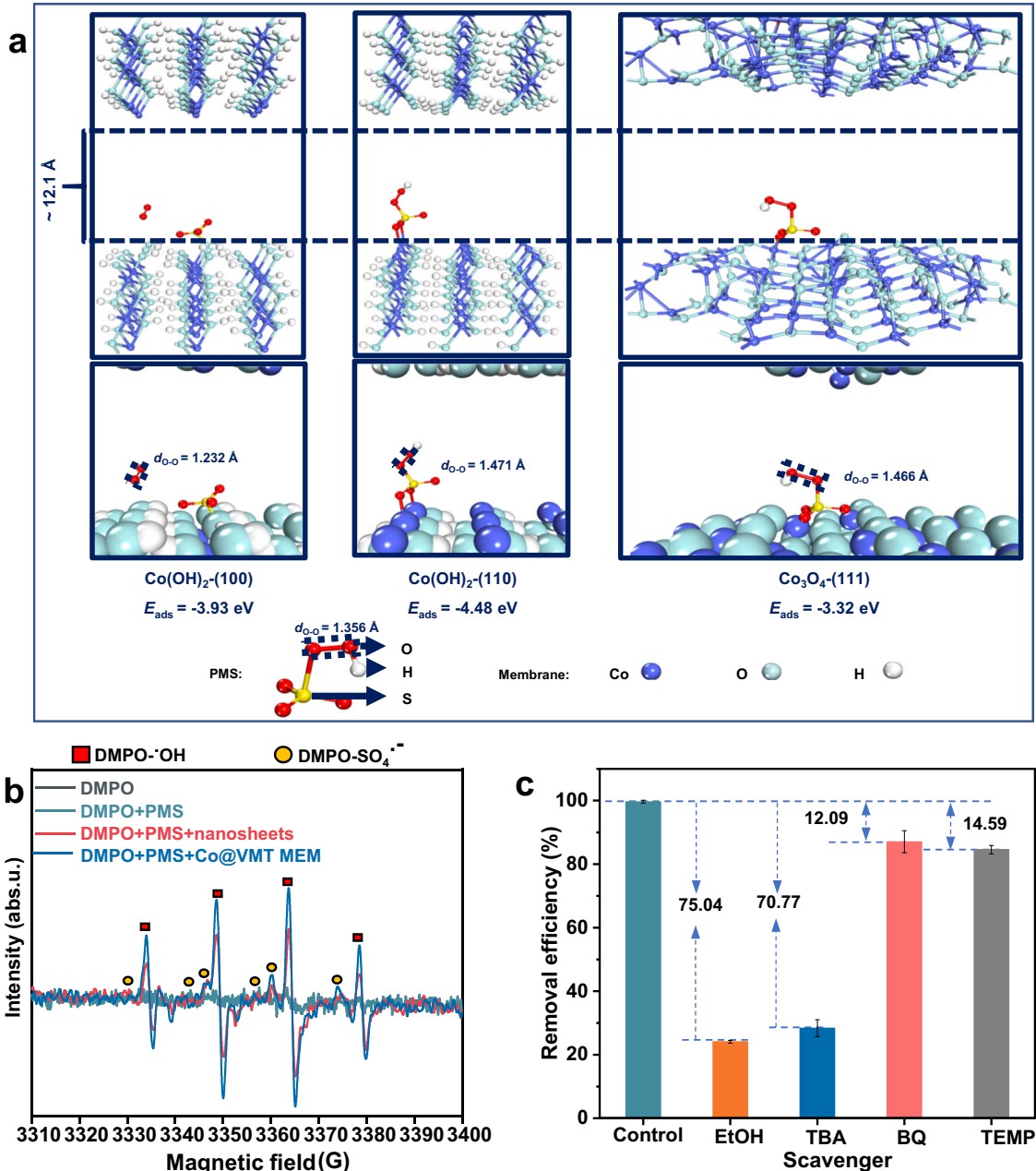

**Fig. 4 | Molecular mechanisms of radical generation and pollutant removal revealed by DFT atomistic simulations and identification of reactive species.** **a** Adsorption free energy ($E_{ads}$) of PMS on the exposed (100) and (110) planes of α-Co(OH)$_2$ and (111) plane of Co$_3$O$_4$ within the intralayer of Co@VMT MEM. **b** EPR spectra of ·OH and SO$_4$·⁻ radicals generated by different catalyst systems. (Reaction conditions: [Ranitidine] = 10 ppm, [PMS] = 20 ppm, [DMPO] = 0.3 mM, pH = 4.0,

$T$ = 298 K, reaction time = 10 min). **c** Comparison of the removal efficiency under different quenching conditions. Reaction conditions: [Ranitidine] = 10 ppm, [PMS] = 20 ppm, [EtOH] = 1 M, [TBA] = 500 mM, [TEMP] = 100 mM, [*p*-BQ] = 5 mM, pH = 4.0, $T$ = 298 K, Pressure = 1 bar). Error bars represent standard deviation of three measurements.

(PMS adsorbed on (111) plane of Co$_3$O$_4$) and 1.471 Å (PMS adsorbed on (110) plane of α-Co(OH)$_2$), indicating the spontaneous dissociation of PMS and its transformation into ROS for organic pollutant degradation[18].

Next, the generation of reactive species in the Co@VMT MEM/PMS system was detected by EPR. 5,5-Dimethyl-1-pyrroline (DMPO) can capture ·OH and SO$_4$·⁻ species. Signals corresponding to both DMPO−·OH adducts (hyperfine coupling constants $A_N = A_H = 14.9$ G) and DMPO−SO$_4$·⁻ adducts ($A_N = 13.2$ G, $A_H = 9.6$ G, $A_H = 1.48$ G and $A_H = 0.78$ G) were detected in the Co@VMT MEM/PMS system (Fig. 4b). SO$_4$·⁻ can react with OH⁻ or H$_2$O to yield ·OH. The relative intensity of DMPO−·OH adducts was much stronger than that of

DMPO−SO$_4$·⁻ adducts and remained stable with the reaction time (Supplementary Fig. 13). It is worth noting that the signal intensities of DMPO−·OH and DMPO−SO$_4$·⁻ generated by the Co@VMT MEM/PMS system were much higher compared with the Co@VMT nanosheets/PMS system. This result indicated that the interlayer/intralayer-confined nanochannels within the membrane can facilitate the full contact of PMS with the Co catalytic active sites to yield more reactive radicals[17,28]. In addition, 2,2,6,6-tetramethyl-4-piperidone hydrochloride (TEMP) and p-benzoquinone (*p*-BQ) were used to trap singlet oxygen (¹O$_2$) and superoxide radical (•O$_2$⁻) species, respectively. The triplet peak signal (1:1:1, $A_N = 16.9$ G) of ¹O$_2$ and the signal of *p*-BQ−•O$_2$⁻ adducts were not detected in either Co@VMT MEM/PMS or Co@VMT

nanosheets/PMS systems during the different time intervals (Supplementary Fig. 14). Subsequent quenching experiments further confirmed the types of reactive species and their contribution to pollutant degradation[17,23]. As shown in Fig. 4c, ethanol (EtOH) served as a radical scavenger for ·OH and $SO_4^{·-}$ species and tert-butyl alcohol (TBA) served to quench ·OH. The presence of EtOH and TBA strongly hindered the ranitidine degradation by 75.04% and 70.77%, respectively, indicating the predominant contribution of ·OH radicals to pollutant degradation. In contrast, much smaller contributions of $^1O_2$ (12.09%) and •$O_2^-$ species (14.59%) were observed, which was in agreement with the EPR results (Supplementary Fig. 14).

Overall, the pollutant degradation process mainly depended on ·OH radicals, and can be described as follows[17]:

$$\equiv Co(II) + HSO_5^- \rightarrow \equiv Co(III) + SO_4^{•-} + OH^- \tag{1}$$

$$SO_4^{•-} + H_2O \rightarrow SO_4^{2-} + {}^{•}OH + H^+ \tag{2}$$

$$\equiv Co(III) + HSO_5^- \rightarrow \equiv Co(II) + SO_5^{•-} + H^+ \tag{3}$$

$$SO_5^{•-} + O^{2-} \rightarrow SO_4^{•-} + O_2 \tag{4}$$

$$Pollutants + {}^{•}OH \rightarrow Degradation\ products/CO_2 + H_2O \tag{5}$$

First, $\equiv Co(II)$ can directly interact with PMS to produce $SO_4^{·-}$ (Eq. (1)). Subsequently, ·OH is generated through hydrolysis of $SO_4^{·-}$ (Eq. (2)). The reduction of $\equiv Co(III)$ to $\equiv Co(II)$ is accompanied by the production of $SO_5^{·-}$ (Eq. (3)). Maintaining PMS activation efficiency requires proper balance between the $\equiv Co(III)$ and $\equiv Co(II)$ states for the long-term, high-efficiency catalytic performance of the Co@VMT MEM/PMS system. A slight conversion (2.13%) of $\equiv Co(II)$ to $\equiv Co(III)$ after 107 h of continuous test was indicated by the XPS analysis, supporting the excellent chemical stability of the Co@VMT MEM system (Supplementary Fig. 15 and Supplementary Table 9). This result was also confirmed by the XANES results that there was a tiny shift for Co to the high positive charge of +3 after reaction (Supplementary Fig. 16a). Note that Fe (with the high positive charge of +3) within VMT unlikely participated in catalytic reactions. Herein, the bulk VMT was an expandable 2:1 mineral with its structure shown in Supplementary Fig. 1. Its crystal lattice consisted of one octahedral sheet sandwiched between two opposing tetrahedral sheets with $Fe^{3+}$ replacing $Mg^{2+}$ in the octahedral sheet by isomorphic substitution[31]. These Fe ions were situated within the central layer of VMT and thus were almost impossible to access PMS for activation, which was confirmed by the XANES results that there was a negligible difference between the XANES spectra at the Fe K-edge of the fresh and used Co@VMT MEM (Supplementary Fig. 16b). As such, the Co nanocatalysts were the critical catalytic active sites. Then, $SO_4^{·-}$ can be generated by the reduction of $SO_5^{·-}$ (Eq. (4)). Eventually, pollutants are predominantly degraded by the reactions with ·OH radicals (Eq. (5)).

The ranitidine degradation by-products and their toxicity were further investigated after the treatment by the Co@VMT MEM/PMS system[17]. Eleven intermediates (P1-P11) were obtained, as shown in Supplementary Fig. 17 and Supplementary Table 10. The main degradation products, except P5 and P7, were not harmful and had substantially lower toxicity than ranitidine. Moreover, the degradation products of P5 and P7 were completely converted into other non-harmful components after 20 min (Supplementary Fig. 18), indicating that safe and non-toxic effluent water quality could be ensured by the Co@VMT MEM/PMS system.

To address the persistent permeability-selectivity trade-off issue of common membrane-based water purification technologies, here we innovatively combined membrane filtration and advanced oxidation processes to develop a 2D nanofluidic catalytic cobalt-functionalized vermiculite (Co@VMT) membrane. The Co@VMT membrane demonstrated the high water permeance of 122.4 L·m$^{-2}$·h$^{-1}$·bar$^{-1}$ and ~100% degradation of diverse organic pollutants with good stability for over 107 h. The achieved water permeance of 122.4 L·m$^{-2}$·h$^{-1}$·bar$^{-1}$ for Co@VMT membrane was two orders of magnitude higher compared to the VMT membrane (1.1 L·m$^{-2}$·h$^{-1}$·bar$^{-1}$). The revealed pollutant removal mechanism of the Co@VMT membrane/PMS system is based on degradation and minimization, which is substantially different from the molecular sieving-based rejection mechanism of VMT membrane with the concentrated pollutants remaining in the brine. In addition, safe and non-toxic effluent water quality was ensured by the Co@VMT MEM/PMS system. Atomistic MD and DFT simulations were performed to elucidate the filtration and catalytic mechanisms. The results of MD simulations confirmed the effective reactions between ROS and target pollutants and the fast transport of water within the nanochannels of Co@VMT membrane. DFT calculations confirmed the spontaneous activation process of PMS on the surface of Co nanocatalysts and mechanisms of ROS generation for organic pollutant degradation. EPR and quenching experiments confirmed the dominant contribution of ·OH radicals for pollutant degradation. This work paves the way towards the development of next-generation nanofluidic catalytic membranes, which can overcome the membrane permeability-selectivity trade-off of the current water filtration and purification technologies.

## Methods

### Preparation of monolayer VMT nanosheets

An ion-exchange method was adopted to exfoliate raw VMT to monolayer VMT[19]. 2 g of the bulk VMT was immersed in 200 mL of a saturated NaCl solution, and stirred at 120 °C under reflux for 24 h. After centrifugation (3960 × g, 5 min) and repeated washing with Milli-Q water, the process was repeated with a 2 M LiCl solution, followed by repeated washing with Milli-Q water. The obtained product was soaked in 200 mL of a 30% hydrogen peroxide ($H_2O_2$) solution, and heated at 120 °C with vigorous stirring. Then, after 10 min of bath sonication and centrifugation (990 × g, 60 min), monolayer 2D VMT nanosheets suspension was prepared.

### Preparation of Co@VMT nanocomposites and Co@VMT membranes

The preparation process of Co@VMT nanocomposites included three stages including hydrolysis of Co(Ac)$_2$, heterogeneous nucleation, and electrostatic interactions[20]. 35 mg of 2D VMT particles were dissolved in EtOH/water (93.75 mL/6.25 mL) with ultrasound for 1 h to form a dispersion solution. 0.6 ml of 0.2 M Co(Ac)$_2$ aqueous solution was added to the VMT/EtOH solution dropwise (Co:VMT = 2:10), followed by heating at 80 °C with stirring for 10 h. The reaction mixture was then cooled down and the composite was collected by centrifugation (3960 × g, 10 min) and washed with Milli-Q water. The hybrid was heated and then re-dispersed to 1 mg.mL$^{-1}$ Co@VMT nanocomposite suspension. This solution was vacuum filtered on PVDF membranes to obtain Co@VMT membranes. The dehydroxylation process of Co@VMT membranes was performed at a temperature of 130 °C for 5 h. Co@VMT membranes with different loadings of Co@VMT nanocomposites (i.e., 0.08, 0.16, 0.32, 0.8, mg.cm$^{-2}$) were prepared to investigate their effect on permeation flux and pollutant removal.

### Evaluation of flux and removal efficiency of organic pollutants

For batch suspension reaction, 0.12 mg Co@VMT nanosheets (same amount as the optimal Co@VMT MEM with Co mass loading of 0.16 mg.cm$^{-2}$) were added into a 150 mL ranitidine solution (10 ppm), and the suspension was stirred for 30 min to achieve an adsorption-desorption equilibrium. The catalytic reaction was then started by

adding OXone (42.10 ppm, 20 ppm PMS) to the suspension and stirring the mixture constantly at 400 rpm. At predetermined intervals throughout the reaction, samples containing 1 mL of the aforementioned solution were obtained and filtered through a cellulose acetate membrane with 0.22 μm pore size, followed by the LC-MS measurement for pollutant removal efficiency evaluation.

The membrane catalytic degradation experiments were carried out on a dead-end filtration unit under an operating pressure of 0.1–0.35 MPa at room temperature. For a typical experiment, a 150 mL solution containing ranitidine or different organic pollutants (10 ppm) and OXone (42.10 ppm, 20 ppm PMS) was filtered through Co@VMT membranes with an effective membrane filtration area of 0.785 cm$^2$, and the permeate samples were collected at regular intervals and analyzed for the quantification using LC-MS. In addition, removal of several other organic pollutants (10 ppm) was assessed to show the general applicability of the Co@VMT/PMS system. The flux of membranes was calculated using the following equation:

$$J = \frac{q}{S * t} \tag{6}$$

where $J$ is the permeate flux (L·m$^{-2}$·h$^{-1}$), q is the membrane permeate volume (L), S is the effective surface area of membrane (m$^2$), t is the time (h).

Removal efficiency ($R$, %) of organic pollutant by the VMT membrane and/or Co@VMT/PMS system can be calculated as follows:

$$R = \frac{C_f - C_p}{C_f} \tag{7}$$

where $C_P$ and $C_f$ are the concentration (ppm) of the organic pollutant in permeate and feed, respectively.

## Data availability
The data that supports the findings of the study are included in the main text and supplementary information files. Source data are provided with this paper.

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

## Acknowledgements

The research was supported by the National Natural Science Foundation of China (52170041), Tsinghua SIGS Start-up Funding (QD2020002N) and Cross-disciplinary Research and Innovation Fund (JC2022006), Key Research and Development Program of Zhejiang Province (2023C03148), the Committee of Science and Technology Innovation of Shenzhen (JCYJ20230807111705011), and Guangdong Higher Education Institutions Innovative Research Team of Urban Water Cycle and Ecological Safety ( 2023KCXTD053 ) . The authors also thank the Anhui Absorption Spectroscopy Analysis Instrument Co, Ltd. for XAFS measurements and analysis.

## Author contributions

Z.Z. conceived the project. M.T. synthesized and characterized the Co@VMT nanosheets and membranes and performed the catalytic activity experiments. Y.L. and S.Z. performed the theoretical calculations. C.Y. performed the XAFS analysis. M.T. and Z.Z. analyzed the results and wrote the manuscript. K.O. commented on the manuscript. Z.Z. revised the manuscript.

## Competing interests

The authors declare no competing interests.

## Additional information

[1]Membrane & Nanotechnology-Enabled Water Treatment Center, Guangdong Provincial Engineering Research Center for Urban Water Recycling and Environmental Safety, Tsinghua Shenzhen International Graduate School, Tsinghua University, Shenzhen 518055 Guangdong, China. [2]School of Environment, Tsinghua University, Beijing 100084, China. [3]National Engineering Laboratory for Vacuum Metallurgy, Kunming University of Science and Technology, Kunming 650093 Yunnan, China. [4]Institute of High Energy Physics, Chinese Academy of Sciences (CAS), Beijing 100049, China. [5]School of Chemistry and Physics, QUT Centre for Materials Science, Queensland University of Technology (QUT), Brisbane, Queensland 4000, Australia.
✉e-mail: zhenghua.zhang@sz.tsinghua.edu.cn

