## [Peer Review File · Nature Communications]

Overcoming the permeability-selectivity challenge in water purification using two-dimensional cobalt-functionalized vermiculite membraneREVIEWER COMMENTS

Reviewer #1 (Remarks to the Author):

Scarcity and inadequate supply of fresh water globally are becoming the huge challenge of the twenty-first century and clean water and sanitation are major global challenges highlighted by the UN Sustainable Development Goals. Water treatment using energy-efficient membrane technologies is one of the most promising solutions. However, the membrane permeability-selectivity trade-off remains the major challenge for synthetic membranes. This manuscript from Prof. Zhang's group from Tsinghua University proposed a two-dimensional cobalt-functionalized vermiculite membrane (Co@VMT), which innovatively combined the properties of membrane filtration and nanoconfinement catalysis and overcame the membrane permeability-selectivity trade-off. The Co@VMT membrane demonstrated a high water flux of $122.4 \text{ L}\cdot\text{m}^{-2}\cdot\text{h}^{-1}\cdot\text{bar}^{-1}$ and $\sim 100\%$ degradation of diverse organic pollutants with good stability for over 107 h. The achieved flux of $122.4 \text{ L}\cdot\text{m}^{-2}\cdot\text{h}^{-1}\cdot\text{bar}^{-1}$ for Co@VMT membrane was two orders of magnitude higher compared to the VMT membrane ($1.1 \text{ L}\cdot\text{m}^{-2}\cdot\text{h}^{-1}\cdot\text{bar}^{-1}$). In addition, different from the molecular sieving-based VMT membrane with the concentrated pollutants remaining in the brine, the Co@VMT membrane/PMS system achieved safe and non-toxic effluent water quality without brine. Molecular dynamics (MD) simulations, density functional theory (DFT) calculations, and state-of-the-art characterizations (e.g. synchrotron X-ray absorption spectroscopy (XAS), electron paramagnetic resonance (EPR)) were adopted to reveal the underlying mechanisms for the breakthrough of membrane permeability-selectivity trade-off by the Co@VMT membrane/PMS system. Overall, this manuscript was well prepared and offered new insights for the development of diverse nanofluidic catalytic membranes to overcome the persistent membrane permeability-selectivity issue of water filtration and purification technologies. As such, this manuscript is recommended for publication. The following issues should be considered to further improve this work.

- 1.Co is the most effective activator for peroxymonosulfate (PMS) to produce reactive oxygen species (ROS). However, according to the EDS mapping results in Figure 1 h, Fe is also high in the Co@VMT membrane and might also contribute to the activation of PMS as Fe is also one of the activators for PMS. However, it seems like the authors did not evaluate the contribution of Fe in the performance part. In the mechanism part, the authors mentioned that Fe (with the high positive charge of +3) within VMT unlikely participated in catalytic reactions. What is the reason for that?
- 2.The authors mentioned that the confinement spaces provided by the intralayer and interlayer paths within the Co@VMT MEM can facilitate the catalytic decomposition of PMS molecules into ROS as well as collisions and reactions between ROS and pollutants (Fig. 3c), resulting in the achieved efficient degradation and mineralization of organic pollutants (Fig. 2). Which one, either intralayer or interlayer paths, plays a more important role in the catalytic process. This should be indicated in the manuscript, similar as the author's previous work (Nat Commun 13, 4010 (2022)).
- 3.Fig 2a, Is there any height difference between Co@VMT and VMT nanosheets. This should be indicated in the manuscript.
- 4.There is a huge difference between the Co@VMT membrane and Co@VMT nanosheets system in terms of the mineralization rate of ranitidine (Figure S9). The related reasons should be supplemented in

the manuscript.

Reviewer #2 (Remarks to the Author):

This paper presents a two-dimensional cobalt-functionalized vermiculite membrane to activate peroxymonosulfate for degradation of several organic pollutants. It is a topic of interest to the researchers in the related areas, and the phenomena presents in this paper are very interesting. However, it seems that this manuscript is still lack of significant novelty compared with the previous studies. I don't recommend the publication of the manuscript in the journal in the current form. My detailed comments are as follows:

1. The introduction section should be refined to better clarify the novelty of this work. Some membranes for activating peroxymonosulfate have been reported previously, such as CuCo layered double hydroxides composite membrane (Chemical Engineering Journal 398 (2020) 125676), ceramic membrane with Co coated graular active carbon (Separation and Purification Technology 291 (2022) 120874), metal-organic framework-derived membrane (Applied Catalysis B: Environmental 312 (2022) 121419). I can not see the advance in the work beyond several other studies already available.
2. In recent years, PMS activation technology has draw great attention in treatment of organic pollutants in wastewater, and large amount of high-efficient catalysts have been prepared, i.e. cobalt cross-linked ordered mesoporous carbon (Chemical Engineering Journal 472 (2023) 145060), LaMnO₃ nanocatalysts (Chemical Engineering Journal 473 (2023) 145343), MOFs-derived Fe-Co-Cu oxycarbide (Chemical Engineering Journal 468 (2023) 143444). In this paper, the cobalt-functionalized vermiculite membrane was fabricated to activate PMS. Whether the catalyst designed in the form of membrane could improved the activation efficiency of PMS worth further discussion.
3. During the process of membrane filtration, the utilization of PMS may be reduced due to the much shorter retention time compared with the traditional PMS activation technology. The water flux could affect the treatment efficiency of the organic pollutants. Suitable flux, but not highest flux, is really needed for the whole treatment system. Thus, it is necessary to discuss the relationship between the flux and the treatment efficiency in this work. In addition, uniform unit of flux should be used.
4. The authors compared the treatment efficiency of Co@VMT MEM/PMS system and the Co@VMT nanosheets/PMS heterogeneous catalysis system. The detailed experimental conditions of the heterogeneous catalysis system should be provided and the experimental conditions should be optimized before.
5. It is suggested that not only the treatment efficiency, but also the cost of the Co@VMT MEM/PMS system and the Co@VMT nanosheets/PMS heterogeneous catalysis system should be compared.
6. Since the membrane size exclusion was not the main contributors for the removal of the organic pollutants, why not improve the membrane flux by simply enlarging the membrane pores?

Reviewer #3 (Remarks to the Author):

In this work, a novel nanofluidic catalytic membrane is developed and investigated in comparison with the state-of-the-art. The membrane characterization was elegantly performed through AFM, TEM, XRD, XPS, XAS, BET and SEM analysis. On the other hand, some important aspects related to the performance evaluation still needs to be clarified.

Specific comments include the following:

1. A material and method section should be introduced to report useful information on the filtration setup and testing protocol such as the membrane area, the initial volume of feed, the final recovery other than specify if a dead end or cross-flow system was used.
2. Fig. 2b is described using the term "removal efficiency" but the figure is expressed with C/C_0 , which makes difficult to be interpreted. I would suggest changing the y-axis of Fig 2b and Fig 2g by using the removal efficiency to make it coherent with other figures and with the discussion.
3. Figures are sometimes too crowded. I would suggest moving less significant figures from the main manuscript to the SI to better highlight the most important ones.
4. Figure 3 shows the interaction between molecules during the degradation process but is not that clear how it is evolving from the "initial state" to "10 ps". I would suggest to better link the figure with the degradation process reported from line 341 to 344.
5. Regarding the degradation process, the proposed reaction path overall releases H^+ which in turn should lower the pH during the filtration process. According to Fig 2h, this effect should be negligible on the removal efficiency as not influenced by the feed pH. However, stabilized flux data in function of the pH should also be showed in the main manuscript as precipitation can occur when dealing at different pH during AOP.
6. The permeability-selectivity trade-off of Fig 2c should be evaluated also for different pH, especially in the neutral region. Moreover, since this membrane is compared with others in Fig 2c, it should be specified the pH at which other methods were investigated to retrieve the first order rate constant.

Overcoming the permeability-selectivity challenge in water purification using two-dimensional cobalt-functionalized vermiculite membrane

REVIEWER 1

General Comment: Scarcity and inadequate supply of fresh water globally are becoming the huge challenge of the twenty-first century and clean water and sanitation are major global challenges highlighted by the UN Sustainable Development Goals. Water treatment using energy-efficient membrane technologies is one of the most promising solutions. However, the membrane permeability-selectivity trade-off remains the major challenge for synthetic membranes. This manuscript from Prof. Zhang's group from Tsinghua University proposed a two-dimensional cobalt-functionalized vermiculite membrane (Co@VMT), which innovatively combined the properties of membrane filtration and nanoconfinement catalysis and overcame the membrane permeability-selectivity trade-off. The Co@VMT membrane demonstrated a high water flux of $122.4 \text{ L}\cdot\text{m}^{-2}\cdot\text{h}^{-1}\cdot\text{bar}^{-1}$ and ~100% degradation of diverse organic pollutants with good stability for over 107 h. The achieved flux of $122.4 \text{ L}\cdot\text{m}^{-2}\cdot\text{h}^{-1}\cdot\text{bar}^{-1}$ for Co@VMT membrane was two orders of magnitude higher compared to the VMT membrane ($1.1 \text{ L}\cdot\text{m}^{-2}\cdot\text{h}^{-1}\cdot\text{bar}^{-1}$). In addition, different from the molecular sieving-based VMT membrane with the concentrated pollutants remaining in the brine, the Co@VMT membrane/PMS system achieved safe and non-toxic effluent water quality without brine. Molecular dynamics (MD) simulations, density functional theory (DFT) calculations, and state-of-the-art characterizations (e.g. synchrotron X-ray absorption spectroscopy (XAS), electron paramagnetic resonance (EPR)) were adopted to reveal the underlying mechanisms

for the breakthrough of membrane permeability-selectivity trade-off by the Co@VMT membrane/PMS system. Overall, this manuscript was well prepared and offered new insights for the development of diverse nanofluidic catalytic membranes to overcome the persistent membrane permeability-selectivity issue of water filtration and purification technologies. As such, this manuscript is recommended for publication. The following issues should be considered to further improve this work.

Response: Thank you for your recognition of this work and your suggestions on this manuscript. A point-by-point response on all issues raised is provided below.

Comment 1: Co is the most effective activator for peroxymonosulfate (PMS) to produce reactive oxygen species (ROS). However, according to the EDS mapping results in Figure 1 h, Fe is also high in the Co@VMT membrane and might also contribute to the activation of PMS as Fe is also one of the activators for PMS. However, it seems like the authors did not evaluate the contribution of Fe in the performance part. In the mechanism part, the authors mentioned that Fe (with the high positive charge of +3) within VMT unlikely participated in catalytic reactions. What is the reason for that?

Response: Thank you for your comments. Herein, the bulk VMT was an expandable 2:1 mineral with its structure shown in Fig. S1. Its crystal lattice consisted of one octahedral sheet sandwiched between two opposing tetrahedral sheets with Fe³⁺ replacing Mg²⁺ in the octahedral sheet by isomorphic substitution (Applied Clay Science 190 (2020) 105543; Proc. Natl. Acad. Sci. USA 96 (1999) 3358-3364). These Fe ions were situated within the central layer of VMT and thus were almost impossible to access PMS for activation, which was confirmed by the XANES results that there was a negligible difference between the XANES spectra at the Fe K-edge of the fresh and used Co@VMT MEM (Fig. S16b). In contrast, Co nanoparticles were positioned on the surfaces of VMT nanosheets and can easily access PMS for activation with the generation of ROS to degrade pollutants.

We have supplemented the related contents to address this issue in the revised manuscript as shown below.

“Note that Fe (with the high positive charge of +3) within VMT unlikely participated in catalytic reactions. Herein, the bulk VMT was an expandable 2:1 mineral with its structure shown in Fig. S1. Its crystal lattice consisted of one octahedral sheet sandwiched between two opposing tetrahedral sheets with Fe^{3+} replacing Mg^{2+} in the octahedral sheet by isomorphic substitution³¹. These Fe ions were situated within the central layer of VMT and thus were almost impossible to access PMS for activation, which was confirmed by the XANES results that there was a negligible difference between the XANES spectra at the Fe K-edge of the fresh and used Co@VMT MEM (Fig. S16b).” (Page 19, Line 377-383)

Figure S1. The macrostructure of bulk VMT. Schematic of VMT bulk structure stacked by adjacent VMT nanosheets through electrostatic attraction with interlayer cations.

Figure S16. Chemical stability evaluation. Normalized XANES spectra at the Co K-edge (a) and Fe K-edge (b) of the fresh and used Co@VMT membrane.

Comment 2: The authors mentioned that the confinement spaces provided by the intralayer and interlayer paths within the Co@VMT MEM can facilitate the catalytic decomposition of PMS molecules into ROS as well as collisions and reactions between ROS and pollutants (Fig. 3c), resulting in the achieved efficient degradation and mineralization of organic pollutants (Fig. 2). Which one, either intralayer or interlayer paths, plays a more important role in the catalytic process. This should be indicated in the manuscript, similar as the author's previous work (Nat Commun 13, 4010 (2022)).

Response: Thank you for the suggestion. We have conducted molecular dynamics (MD) simulations to illustrate the transport properties of PMS, ranitidine, and water within the Co@VMT membranes (Fig 3a). As shown in Fig 3a and Supplementary Video 1, the narrow interlayer free spacing (3.09 Å) of VMT MEM could barely allow the mass transfer of water, ranitidine (0.570 × 0.460 × 1.68 nm), and PMS (0.315 × 0.305 × 0.350 nm) (Fig. S12) within interlayer paths, which resulted in the increased mass transfer resistance and a low flux (Fig. 2a) with intralayers as the main mass transfer paths. However, the enlarged interlayer free spacing (4.76 Å) of the

Co@VMT MEM not only enabled the transport of water and PMS molecules through the interlayer paths, but also boosted the mass transfer through the downward intralayer paths with a free spacing of 1.21 nm working as the predominant mass transport channel for large ranitidine molecules (Supplementary Video 2). Therefore, a high flux was achieved, two orders of magnitude higher than that of the VMT MEM (Fig. 2a). The Mean Square Displacement (MSD) curves (Fig. 3b) displayed a significant increase in the mass transfer rate through Co@VMT MEM compared with VMT MEM, especially for water and PMS molecules. Water exhibited the fastest mass transfer in nanochannels, which may be attributed to high capillary pressures and the highly ordered structure of water in nano-spaces. The above MD results indicate that PMS could swiftly interact with the Co catalytic sites in both the interlayer and intralayer paths of Co@VMT MEM (Fig. 3a), contributing to the efficiently catalytic decomposition of PMS molecules into ROS as well as collisions and reactions between ROS and pollutants (Fig. 3c). Moreover, the confinement spaces provided by the intralayer and interlayer paths within the Co@VMT MEM can significantly reduce the migration distance between ROS and pollutants and thus remarkably facilitate the utilization of ROS (Fig. 3c), resulting in the achieved efficient degradation and mineralization of organic pollutants (Fig. 2).

Figure 2 | Membrane permeability-selectivity performance evaluation of the Co@VMT membrane/PMS system. (a) Flux and ranitidine removal efficiency of VMT membrane and Co@VMT membrane/PMS systems.

Figure 3 | MD simulations of mass transfer within VMT and Co@VMT membranes. (a) MD simulations of the diffusion of PMS, ranitidine, and H₂O molecules within VMT and Co@VMT membranes; (b) MSD curves of PMS, ranitidine, and H₂O molecules within VMT and Co@VMT membranes.

Figure S12. Molecular size analysis. The molecular size of PMS and ranitidine molecules.

We have supplemented these contents in the revised manuscript as shown below.

“As shown in Fig 3a and Supplementary Video 1, the narrow interlayer free spacing (3.09 Å) of VMT MEM could barely allow the mass transfer of water, ranitidine ($0.570 \times 0.460 \times 1.68$ nm), and PMS ($0.315 \times 0.305 \times 0.350$ nm) (Fig. S12) within interlayer paths, which resulted in the increased mass transfer resistance and a low flux (Fig. 2a) with intralayers as the main mass transfer paths^{15, 27}. However, the enlarged interlayer free spacing (4.76 Å) of the Co@VMT MEM not only enabled the transport of water and PMS molecules through the interlayer paths, but also boosted the mass transfer through the downward intralayer paths with a free spacing of 1.21 nm working as the predominant mass transport channel for large ranitidine molecules (Supplementary Video 2). Therefore, a high flux was achieved, two orders of magnitude higher than that of the VMT MEM (Fig. 2a). The Mean Square Displacement (MSD) curves (Fig. 3b) displayed a significant increase in the mass transfer rate through Co@VMT MEM compared with VMT MEM, especially for water and PMS molecules. Water exhibited the fastest mass transfer in nanochannels, which may be attributed to high capillary pressures and the highly ordered structure of water in nano-spaces^{17, 29, 30}. The above MD results indicate that PMS could swiftly interact with the Co catalytic sites in both the interlayer and intralayer paths of Co@VMT MEM (Fig. 3a), contributing to the efficiently catalytic decomposition of PMS molecules into ROS as well as collisions and reactions between ROS and pollutants (Fig. 3c). Moreover, the confinement spaces provided by the intralayer and interlayer paths within the Co@VMT MEM can significantly reduce the migration distance between ROS and pollutants and thus remarkably facilitate the utilization of ROS (Fig. 3c)¹⁷, resulting in the achieved efficient degradation and mineralization of organic pollutants (Fig. 2).” (Page 15, Line 294-314)

Comment 3: Fig 2a, Is there any height difference between Co@VMT and VMT nanosheets. This should be indicated in the manuscript.

Response: This is a good suggestion. Thanks. The thickness of Co@VMT nanosheets was 1.04 nm (Fig. S2b), which was almost the same to that of VMT nanosheets (1.02 nm) (Fig. 1a). This has been supplemented in the revised manuscript as shown below.

“The thickness of Co@VMT nanosheets was 1.04 nm (Fig. S2b), which was almost the same to that of VMT nanosheets (1.02 nm) (Fig. 1a).” (Page 6, Line 120-122)

Figure S2. AFM (b) characterization. AFM image with the height profile of Co@VMT nanosheets.

Comment 4: There is a huge difference between the Co@VMT membrane and Co@VMT nanosheets system in terms of the mineralization rate of ranitidine (Figure S9). The related reasons should be supplemented in the manuscript.

Response: We appreciate for your valuable suggestion. The reasons for the high mineralization rate of ranitidine by the Co@VMT MEM/PMS system include the more efficient PMS decomposition, more generated ROS, and abundant nanochannels-confined AOP spaces where electronic interactions, enrichment of ROS

and catalytic reactions were triggered to achieve the efficient removal of pollutants. The detailed reasons have been discussed in the mechanism part. Herein, we have supplemented the related contents in the degradation part to address this issue as shown below.

“In addition, the PMS decomposition efficiency for the Co@VMT MEM/PMS system (75.6%) was also significantly higher compared to the other two systems (37.54% for Co@VMT nanosheets/PMS system and 11.23% for PMS only system) (Fig. S10). This result demonstrated the high catalytic efficiency of the Co@VMT MEM/PMS system to activate PMS for ROS production²³. Moreover, the interlayers (4.76 Å) and intralayers (1.21 nm) of Co@VMT MEM could offer abundant AOP confinement spaces, wherein the migration distances between ROS and pollutants were greatly shorten and the collisions and reactions between ROS and pollutants were significantly strengthened. As such, a significantly high ranitidine mineralization rate was achieved by the Co@VMT MEM/PMS system with the more detailed mechanisms elucidated in the following part.” (Page 12, Line 230-239)

REVIEWER 2

This paper presents a two-dimensional cobalt-functionalized vermiculite membrane to activate peroxymonosulfate for degradation of several organic pollutants. It is a topic of interest to the researchers in the related areas, and the phenomena presents in this paper are very interesting. However, it seems that this manuscript is still lack of significant novelty compared with the previous studies. I don't recommend the publication of the manuscript in the journal in the current form. My detailed comments are as follows:

Response: Thank you for your recognition of this work and your suggestions on this manuscript. A point-by-point response on all issues raised is provided below.

Comment 1: The introduction section should be refined to better clarify the novelty of this work. Some membranes for activating peroxydisulfate have been reported previously, such as CuCo layered double hydroxides composite membrane (Chemical Engineering Journal 398 (2020) 125676), ceramic membrane with Co coated granular active carbon (Separation and Purification Technology 291 (2022) 120874), metal-organic framework-derived membrane (Applied Catalysis B: Environmental 312 (2022) 121419). I can not see the advance in the work beyond several other studies already available.

Response: We appreciate for your comments. We have carefully and thoroughly read the recommend works. These three works investigated the organic pollutants removal performance by PMS activation using CuCo LDH membrane, Co coated ceramic membrane and MOF-derived membrane. However, none of these works are related to 2D laminar membranes as well as the membrane permeability-selectivity trade-off. The main knowledge gap of this work is to overcome the membrane permeability-selectivity trade-off using 2D laminar membranes. We have clearly pointed out the knowledge gap of this work in the introduction part as shown below.

“However, the membrane permeability-selectivity trade-off is still the major hurdle for commercial applications of synthetic membranes, wherein permeability limits the flow rate while selectivity affects the products of the separation process⁴.” (Page 3, Line 53-56)

“Two-dimensional (2D) materials have unique physicochemical properties, atomic thickness, large aspect ratio and chemical flexibility. This is why many 2D materials such as graphene, graphene oxide (GO), MXene, molybdenum disulfide (MoS₂) and boron nitride (BN), have been recently employed for the fabrication of 2D materials-based membranes to overcome the permeability-selectivity trade-off⁵. There are two kinds of 2D materials-based membranes: (i) laminar/lamellar membranes with interlayer/intralayer galleries synthesized by stacking 2D nanosheets, and (ii) nanoporous membranes....” (Page 3, Line 58-64)

“However, the difficulty in construction of uniform and well-dispersed nanopores and the high economic cost of drilling restrict the development of 2D porous membranes^{6, 9}. In addition, 2D organic framework materials (e.g., COF, MOF) lack robust mechanical strength to form large surface area membranes. These thin-film materials also struggle with the existing intrinsic or extrinsic defects (e.g., grain boundaries) between single crystalline units, leading to non-selective transport^{12, 13}.” (Page 4, Line 71-75)

“Water flux through narrow nanochannels in 2D laminar membranes is low and requires extra spacer species to improve¹⁵. The irregular stacking of 2D nanosheets with microporous defects also limits the water-solute selectivity of 2D laminar membranes¹⁴⁻¹⁶. Furthermore, only a few reliable techniques for exfoliating high aspect ratio and intact 2D monolayers from bulk crystals are currently available, while the existing synthesis methods are not environment-friendly and involve strong acids or bases, cumbersome oxidation and reduction processes^{9, 10}. Collectively, these factors pose major challenges for the development and applications of 2D laminar membranes.” (Page 4, Line 81-88)

“To address these challenges, here we report on a novel approach to overcome the membrane permeability-selectivity trade-off using a nanoconfinement catalytic process enabled by the judiciously designed 2D cobalt-functionalized vermiculite (Co@VMT) membrane. Cobalt was chosen as the most effective activator for peroxymonosulfate (PMS) to produce reactive oxygen species (ROS). The proposed membrane-based nanoconfined heterogeneous catalysis approach relies on our proprietary multi-functional membrane that can sustain both the advanced oxidation processes (AOPs) and membrane filtration. The catalysts loaded in the interlayer and/or intralayer of the 2D laminar membrane improved the water flux by increasing the spacing of the interlayer and/or intralayer, while simultaneously ensuring the effective degradation and mineralization of organic pollutants^{17, 18}.” (Page 5, Line 89-97)

Comment 2: In recent years, PMS activation technology has draw great attention in treatment of organic pollutants in wastewater, and large amount of high-efficient catalysts have been prepared, i.e. cobalt cross-linked ordered mesoporous carbon (Chemical Engineering Journal 472 (2023) 145060), LaMnO₃ nanocatalysts (Chemical Engineering Journal 473 (2023) 145343), MOFs-derived Fe-Co-Cu oxycarbide (Chemical Engineering Journal 468 (2023) 143444). In this paper, the cobalt-functionalized vermiculite membrane was fabricated to activate PMS. Whether the catalyst designed in the form of membrane could improved the activation efficiency of PMS worth further discussion.

Response: Thanks for your comments. We have carefully and thoroughly read the recommend three works, which are about the synthesis and employment of new nanocatalysts for heterogeneous catalytic reactions in batch suspension solutions. However, none of these works are related to membrane, especially for membrane-based nanoconfinement catalysis. Our team has been working to develop membrane-based nanoconfinement catalysis systems for efficient degradation of pollutants, and has published a few dozen papers in this area (Nat. Commun. 2022, 13, 4010; Appl. Catal. B. 2023, 322, 122098; 2021, 284, 119720; Chem Catal. 2022, 2,1-13; Water Res. 2023, 230, 119577; Chem. Eng. J, 2023, 463, 142340; 2022, 449, 137811; 2022, 443, 136495; 2022, 435, 135126; J. Mater. Chem. A ; 2023, 11, 18933; 2021, 9, 19817-19833; J. Hazard. Mater. 2022, 425, 127988 et. al.). The improved decomposition of PMS by nanoconfined membrane compared to batch suspension solutions could be due to the significantly shortened migration distances between PMS and catalysts, more collisions and reactions between PMS and catalysts within the interlayer/intralayer-confined nanochannels within the membrane. This is indeed the case herein. As shown in Fig. S10, the PMS decomposition efficiency for the Co@VMT MEM/PMS system (75.6%) was also significantly higher compared to the other two systems (37.54% for Co@VMT nanosheets/PMS system and 11.23% for PMS only system). This result demonstrated the high catalytic efficiency of the

Co@VMT MEM/PMS system to activate PMS for ROS production. We have pointed out this point in the revised manuscript and also supplemented related discussion as shown below.

“In addition, the PMS decomposition efficiency for the Co@VMT MEM/PMS system (75.6%) was also significantly higher compared to the other two systems (37.54% for Co@VMT nanosheets/PMS system and 11.23% for PMS only system) (Fig. S10). This result demonstrated the high catalytic efficiency of the Co@VMT MEM/PMS system to activate PMS for ROS production²³. Moreover, the interlayers (4.76 Å) and intralayers (1.21 nm) of Co@VMT MEM could offer abundant AOP confinement spaces, wherein the migration distances between ROS and pollutants were greatly shorten and the collisions and reactions between ROS and pollutants were significantly strengthened. As such, a significantly high ranitidine mineralization rate was achieved by the Co@VMT MEM/PMS system with the more detailed mechanisms elucidated in the following part.” (Page 12, Line 230-290)

Figure S10. PMS decomposition performance. Decomposition efficiency of PMS in different systems.

Comment 3: During the process of membrane filtration, the utilization of PMS may be reduced due to the much shorter retention time compared with the traditional PMS activation technology. The water flux could affect the treatment efficiency of the organic pollutants. Suitable flux, but not highest flux, is really needed for the whole treatment system. Thus, it is necessary to discuss the relationship between the flux and the treatment efficiency in this work. In addition, uniform unit of flux should be used.

Response: This is a very good suggestion. We agree with the reviewer that suitable flux is necessitated to ensure high removal efficiency of pollutants. We have supplemented the related discussion about the effect of flux on pollutant removal efficiency in the revised manuscript as shown below. Meanwhile, the uniform unit of $\text{L}\cdot\text{m}^{-2}\cdot\text{h}^{-1}$ was used for flux.

“On the other hand, increasing the membrane water flux shortened the retention time of pollutants in the membrane, potentially diminishing the removal efficiency of pollutants. The removal efficiency of ranitidine remained stable at ~100% when the water flux increased from 122.4 to 258.9 $\text{L}\cdot\text{m}^{-2}\cdot\text{h}^{-1}$ (Fig. 2i). However, the further increase of water flux resulted in the decrease of removal efficiency. When the water flux increased to 342.1 $\text{L}\cdot\text{m}^{-2}\cdot\text{h}^{-1}$, the removal efficiency of ranitidine decreased to 97.8% (Fig. 2i). As such, the membrane water flux should be less than 258.9 $\text{L}\cdot\text{m}^{-2}\cdot\text{h}^{-1}$ in order to achieve a 100% removal efficiency of ranitidine.” (Page 14, Line 275-281)

Comment 4: The authors compared the treatment efficiency of Co@VMT MEM/PMS system and the Co@VMT nanosheets/PMS heterogeneous catalysis system. The detailed experimental conditions of the heterogeneous catalysis system should be provided and the experimental conditions should be optimized before.

Response: Thank you for your suggestion. We have moved the detailed experimental conditions about the fabrication of monolayer VMT nanosheets, Co@VMT nanocomposites and Co@VMT membranes as well as the evaluation of flux and

removal efficiency of organic pollutants from SI to the revised manuscript as shown below. Note that the same amount of catalysts was used for both the batch suspension solution and membrane.

“Evaluation of flux and removal efficiency of organic pollutants

For batch suspension reaction, 0.12 mg Co@VMT nanosheets (same amount as the optimal Co@VMT MEM with Co mass loading of 0.16 mg/cm²) were added into a 150 mL ranitidine solution (10 ppm), and the suspension was stirred for 30 min to achieve an adsorption-desorption equilibrium. The catalytic reaction was then started by adding OXone (42.10 ppm, 20 ppm PMS) to the suspension and stirring the mixture constantly at 400 rpm. At predetermined intervals throughout the reaction, samples containing 1 mL of the aforementioned solution were obtained and filtered through a cellulose acetate membrane with 0.22 μm pore size, followed by the LC-MS measurement for pollutant removal efficiency evaluation.

The membrane catalytic degradation experiments were carried out on a dead-end filtration unit under an operating pressure of 0.1–0.35 MPa at room temperature. For a typical experiment, a 150 mL solution containing ranitidine or different organic pollutants (10 ppm) and OXone (42.10 ppm, 20 ppm PMS) was filtered through Co@VMT membranes with an effective membrane filtration area of 0.785 cm², and the permeate samples were collected at regular intervals and analyzed for the quantification using LC-MS. In addition, removal of several other organic pollutants (10 ppm) was assessed to show the general applicability of the Co@VMT/PMS system. The flux of membranes was calculated using the following equation:

$$J = \frac{q}{S \cdot t} \quad (6)$$

where J is the permeate flux (L·m⁻²·h⁻¹), q is the membrane permeate volume (L), S is the effective surface area of membrane (m²), t is the time (h).

Removal efficiency (R, %) of organic pollutant by the VMT membrane and/or Co@VMT/PMS system can be calculated as follows:

$$R = \frac{C_f - C_p}{C_f} \quad (7)$$

where C_p and C_f are the concentration (ppm) of the organic pollutant in permeate and feed, respectively.” (Page 23-24, Line 23-24)

Meanwhile, Co@VMT membranes with different mass ratios (Co:VMT=1:10, 2:10, 4:10) and different loadings of Co@VMT nanocomposites (i.e., 0.08, 0.16, 0.32, 0.8, mg/cm^2) were prepared to investigate their effect on permeation flux and pollutant removal (Fig. S7). The optimal Co@VMT MEM with Co mass loading of 0.16 mg/cm^2 was chosen for detailed mechanism analysis.

Figure S7. Membrane permeability-selectivity performance evaluation. Flux (a) and removal efficiency of ranitidine (b) for VMT and Co@VMT membranes with different mass loading of Co:VMT (10 ppm ranitidine; 1 bar pressure).

Comment 5: It is suggested that not only the treatment efficiency, but also the cost of the Co@VMT MEM/PMS system and the Co@VMT nanosheets/PMS heterogeneous catalysis system should be compared.

Response: We appreciate for your suggestion. Assembling nanosheets into membrane forms could address the issues like catalyst agglomeration, recycling and reuse during heterogeneous catalytic reactions in batch suspension solutions. Moreover,

membrane-based nanoconfinement catalysis achieved more efficient pollutant degradation performance compared to the heterogeneous catalytic reactions in batch suspension solutions. As such, the Co@VMT MEM/PMS system would be more cost-effective compared to the Co@VMT nanosheets/PMS system. However, the detailed cost calculation is out of the scope of this study. We have supplemented the related contents in the revised manuscript to address this issue as shown below.

“Moreover, assembling nanosheets into membrane forms could address the issues like catalyst agglomeration, recycling and reuse during heterogeneous catalytic reactions in batch suspension solutions. As such, the Co@VMT MEM/PMS system would be more cost-effective and was preferred compared to the Co@VMT nanosheets/PMS system.” (Page 11, Line 221-224)

Comment 6: Since the membrane size exclusion was not the main contributors for the removal of the organic pollutants, why not improve the membrane flux by simply enlarging the membrane pores?

Response: Thanks for your comment. The membrane permeability-selectivity trade-off is still the major hurdle for commercial applications of synthetic membranes, wherein permeability limits the flow rate while selectivity affects the products of the separation process. If we enlarge the membrane pore size to enhance the membrane flux, however, we will sacrifice the membrane selectivity performance. That is why we introduce a novel approach to overcome the membrane permeability-selectivity trade-off using a nanoconfinement catalytic process enabled by the judiciously designed 2D cobalt-functionalized vermiculite (Co@VMT) membrane. The proposed membrane-based nanoconfined heterogeneous catalysis approach relies on our proprietary multi-functional membrane that can sustain both the advanced oxidation processes (AOPs) and membrane filtration. The catalysts loaded in the interlayer and/or intralayer of the 2D laminar membrane improved the water flux by increasing the spacing of the interlayer and/or intralayer, while simultaneously ensuring the

effective degradation and mineralization of organic pollutants. We have indicated this in the revised manuscript as shown below.

“However, the membrane permeability-selectivity trade-off is still the major hurdle for commercial applications of synthetic membranes, wherein permeability limits the flow rate while selectivity affects the products of the separation process⁴.” (Page 3, Line 53-56)

“To address these challenges, here we report on a novel approach to overcome the membrane permeability-selectivity trade-off using a nanoconfinement catalytic process enabled by the judiciously designed 2D cobalt-functionalized vermiculite (Co@VMT) membrane. Cobalt was chosen as the most effective activator for peroxymonosulfate (PMS) to produce reactive oxygen species (ROS). The proposed membrane-based nanoconfined heterogeneous catalysis approach relies on our proprietary multi-functional membrane that can sustain both the advanced oxidation processes (AOPs) and membrane filtration. The catalysts loaded in the interlayer and/or intralayer of the 2D laminar membrane improved the water flux by increasing the spacing of the interlayer and/or intralayer, while simultaneously ensuring the effective degradation and mineralization of organic pollutants^{17, 18}.” (Page 4-5, Line 89-97)

“The revealed pollutant removal mechanism of the Co@VMT membrane/PMS system is based on degradation and minimization, which is substantially different from the molecular sieving-based rejection mechanism of VMT membrane with the concentrated pollutants remaining in the brine.” (Page 21, Line 409-412)

REVIEW 3

In this work, a novel nanofluidic catalytic membrane is developed and investigated in comparison with the state-of-the-art. The membrane characterization was elegantly performed through AFM, TEM, XRD, XPS, XAS, BET and SEM analysis. On the other hand, some important aspects related to the performance evaluation still needs to be clarified. Specific comments include the following:

Response: Thank you for your recognition of this work and your suggestions on this manuscript. A point-by-point response on all issues raised is provided below.

Comment 1: A material and method section should be introduced to report useful information on the filtration setup and testing protocol such as the membrane area, the initial volume of feed, the final recovery other than specify if a dead end or cross-flow system was used.

Response: We appreciate for your valuable suggestion. All these details have been moved from SI to the revised manuscript and we have supplemented one Methods part in the revised manuscript as shown below.

“Evaluation of flux and removal efficiency of organic pollutants

For batch suspension reaction, 0.12 mg Co@VMT nanosheets (same amount as the optimal Co@VMT MEM with Co mass loading of 0.16 mg/cm²) were added into a 150 mL ranitidine solution (10 ppm), and the suspension was stirred for 30 min to achieve an adsorption-desorption equilibrium. The catalytic reaction was then started by adding OXone (42.10 ppm, 20 ppm PMS) to the suspension and stirring the mixture constantly at 400 rpm. At predetermined intervals throughout the reaction, samples containing 1 mL of the aforementioned solution were obtained and filtered through a cellulose acetate membrane with 0.22 μm pore size, followed by the LC-MS measurement for pollutant removal efficiency evaluation.

The membrane catalytic degradation experiments were carried out on a dead-end

filtration unit under an operating pressure of 0.1–0.35 MPa at room temperature. For a typical experiment, a 150 mL solution containing ranitidine or different organic pollutants (10 ppm) and OXone (42.10 ppm, 20 ppm PMS) was filtered through Co@VMT membranes with an effective membrane filtration area of 0.785 cm², and the permeate samples were collected at regular intervals and analyzed for the quantification using LC-MS. In addition, removal of several other organic pollutants (10 ppm) was assessed to show the general applicability of the Co@VMT/PMS system. The flux of membranes was calculated using the following equation:

$$J = \frac{q}{S \cdot t} \quad (6)$$

where J is the permeate flux ($L \cdot m^{-2} \cdot h^{-1}$), q is the membrane permeate volume (L), S is the effective surface area of membrane (m^2), t is the time (h).

Removal efficiency (R , %) of organic pollutant by the VMT membrane and/or Co@VMT/PMS system can be calculated as follows:

$$R = \frac{C_f - C_p}{C_f} \quad (7)$$

where C_p and C_f are the concentration (ppm) of the organic pollutant in permeate and feed, respectively.” (Page 23-24, Line 23-24)

Comment 2: Fig. 2b is described using the term “removal efficiency” but the figure is expressed with C/C_0 , which makes difficult to be interpreted. I would suggest changing the y-axes of Fig 2b and Fig 2g by using the removal efficiency to make it coherent with other figures and with the discussion.

Response: Thanks for your suggestion. As suggested, we have revised the y-axes of Fig 2b and Fig 2g by using the removal efficiency as shown below.

Figure 2 | Membrane permeability-selectivity performance evaluation of the Co@VMT membrane/PMS system. (a) Flux and ranitidine removal efficiency of VMT membrane and Co@VMT membrane/PMS systems; (b) Ranitidine removal efficiency in different catalytic systems; (c) Comparison of the first-order rate constants (k) of different systems. Details in Table S5; (d) Stability test of flux and removal efficiency with operation duration; (e) Removal of different organic pollutants by Co@VMT membrane/PMS system; (f) Comparison of flux and pollutant removal efficiency between Co@VMT membrane/PMS system and previously developed membranes. Details in Table S7; Effects of (g) real water, (h) solution pH and (i) applied pressure on ranitidine removal by Co@VMT membrane/PMS system. (Conditions of the feed solution for all membrane tests: [Ranitidine] or [Other pollutants] = 10 ppm, [PMS] = 20 ppm, pH = 4.0, Pressure = 1 bar, T = 298 K).

Comment 3: Figures are sometimes too crowded. I would suggest moving less significant figures from the main manuscript to the SI to better highlight the most important ones.

Response: We appreciate for your comment. We have 18 figures in the SI and keep 4 important and combined figures in the manuscript. Fig. 1 is about synthesis and microanalysis of Co@VMT membrane and Fig. 2 is about membrane permeability-selectivity performance evaluation of the Co@VMT membrane/PMS system, which the reviewer might think are somehow crowded. However, in order to have a better logic of this work and facilitate the understanding for readers, we decide to keep the 4 important and combined figures in the manuscript after discussion. Taking Fig. 1 as an example, Fig. 1a-d is about the microanalysis of Co@VMT nanosheets and Fig. 1e-l is about the microanalysis of Co@VMT membrane. Fig. 1a-d show the single layer of Co@VMT nanosheets (Fig. 1a-b) and the particle size and distribution of Co nanocatalysts on the VMT surface (Fig. 1c-d). Fig. 1e-l show the interlayer free spacing of Co@VMT membrane (Fig. 1e), lattice plane of α -Co(OH)₂ and lattice plane of Co₃O₄ (Fig. 1f), surface (Fig. 1g) and cross-section and EDS mapping (Fig. 1h) of Co@VMT membrane, chemical state of Co nanocatalysts (Fig. 1i-XPS, Fig. 1j-k-XAS) and membrane pore size (Fig. 1l). These figures combined as Fig. 1 are all the most important ones for the microanalysis of Co@VMT nanosheets and membrane. In contrast, Fig. S1-S6 in the SI are all the supporting figures for Fig. 1. As such, we think the current combined figures are more reasonable to show the better logic of this work and facilitate the understanding for readers.

Comment 4: Figure 3 shows the interaction between molecules during the degradation process but is not that clear how it is evolving from the “initial state” to “10 ps”. I would suggest to better link the figure with the degradation process reported from line 341 to 344.

Response: We appreciate for this great suggestion. The detailed diffusion processes of PMS, ranitidine, and H₂O molecules within VMT and Co@VMT membranes are shown in Supplementary Videos 1 and 2 and Fig. 3 with initial, 7 and 10 ps states. As suggested, we have supplemented the related contents about the potential relation of mass transfer with the degradation process in the revised manuscript as shown below.

“The above MD results indicate that PMS could swiftly interact with the Co catalytic sites in both the interlayer and intralayer paths of Co@VMT MEM (Fig. 3a), contributing to the efficiently catalytic decomposition of PMS molecules into ROS as well as collisions and reactions between ROS and pollutants (Fig. 3c). Moreover, the confinement spaces provided by the intralayer and interlayer paths within the Co@VMT MEM can significantly reduce the migration distances between ROS and pollutants and thus remarkably facilitate the utilization of ROS (Fig. 3c)¹⁷, resulting in the achieved efficient degradation and mineralization of organic pollutants (Fig. 2).”
(Page 15-16, Line 311-318)

Comment 5: Regarding the degradation process, the proposed reaction path overall releases H⁺ which in turn should lower the pH during the filtration process. According to Fig 2h, this effect should be negligible on the removal efficiency as not influenced by the feed pH. However, stabilized flux data in function of the pH should also be showed in the main manuscript as precipitation can occur when dealing at different pH during AOP.

Response: Thank you for your suggestion. In fact, the pH of the feed solution remained stable during the degradation process as shown below in Fig. R1. We also measured the flux at different pHs. As shown in Fig. 2h, the membrane water flux was also stable with a steady water permeance of 122.4 L·m⁻²·h⁻¹·bar⁻¹ and there were no precipitates observed when the solution pH was raised from 6 to 9. We have supplemented the related contents to address this issue in the revised manuscript as shown below.

“When the solution pH was raised from 6 to 9, the degradation efficiency of ranitidine remained at 100% and a steady water permeance of $122.4 \text{ L}\cdot\text{m}^{-2}\cdot\text{h}^{-1}\cdot\text{bar}^{-1}$ was maintained, demonstrating the remarkable adaptability of the Co@VMT MEM/PMS system toward pH variation without precipitation occurred.” (Page 13, Line 268-271)

Figure R1. pH of feed solution. pH of the feed solution over time during the degradation process. (Conditions of the feed solution for the test: [Ranitidine] = 10 ppm, [PMS] = 20 ppm, pH = 4.0, Pressure = 1 bar, T = 298 K).

Figure 2 | Membrane permeability-selectivity performance evaluation of the Co@VMT membrane/PMS system. (h) Effects of solution pH on ranitidine removal by Co@VMT membrane/PMS system.

Comment 6: The permeability-selectivity trade-off of Fig 2c should be evaluated also for different pH, especially in the neutral region. Moreover, since this membrane is compared with others in Fig 2c, it should be specified the pH at which other methods were investigated to retrieve the first order rate constant.

Response: Thank you for your suggestion. We have demonstrated that our Co@VMT membrane/PMS system could break the membrane permeability-selectivity trade-off even when the solution pH was raised from 6 to 9 as shown in Fig. 2h. We have supplemented the related contents to address this issue in the revised manuscript as shown below.

“When the solution pH was raised from 6 to 9, the degradation efficiency of ranitidine remained at 100% and a steady water permeance of $122.4 \text{ L}\cdot\text{m}^{-2}\cdot\text{h}^{-1}\cdot\text{bar}^{-1}$ was maintained, demonstrating the remarkable adaptability of the Co@VMT MEM/PMS system toward pH variation without precipitation occurred.” (Page 13, Line 268-271)

As suggested, we have supplemented the pH for references as shown in Table S5.

Table S5. Comparison of ranitidine removal efficiency over different technologies

No.	Methods	Materials	Ranitidine concentration (mg/L)	Ranitidine removal efficiency	Catalysts dosage	Feed solution pH	Reaction time	$k \text{ (min}^{-1}\text{)}$	Ref
1	Photocatalysis	MoS ₂ /RGO	10	74%	1 g/L	6.5	60 min	0.0208	4
2	Photocatalysis	MoS ₂	10	33%	1 g/L	6.5	60 min	0.00599	4
3	Photocatalysis	RGO	10	35%	1 g/L	6.5	60 min	0.00644	4
4	Photocatalysis	MXene-Ti ₃ C ₂ /MoS ₂	10	88.40%	1 g/L	6.5	60 min	0.03148	5
5	Photocatalysis	MXene-Ti ₃ C ₂	10	18.40%	1 g/L	6.5	60 min	0.0032	5
6	Photocatalysis	TiO ₂	10	100%	0.2 g/L	6.5	45 min	0.146	5

7	Photo-Fenton	TiO ₂ +Fe ²⁺ /H ₂ O ₂	10	100%	0.2 g/L	6.6	22 min	0.23	6
8	Photocatalysis	TiO ₂ nanofiber	3	95%	--	6.5	120 min	0.008	7
9	Photocatalysis	Degussa P25	3	96%	--	6.5	120 min	0.011	7
10	UV photolysis	NH ₂ Cl	5	89.40%	0.051-0.3 g/L	6.0-8.0	5 min	0.33	8
11	Heterogeneous catalysis	Co ₃ O ₄ NS	5	47.20%	0.02 g/L	9.0	30 min	0.021	9
13	Heterogeneous catalysis	OM-Co ₃ O ₄	10	99.20%	0.25 g/L	3.0	7 min	0.719	10
14	Heterogeneous catalysis	nZVIPS@Ti ₃ C ₂	10	92.30%	0.1 g/L	3.0	6 min	0.4311	11
15	Heterogeneous catalysis	BN-Co ₃ O ₄	10	99.60%	0.03 g/L	6.0	10 min	0.682	12
16	Nanoconfinement catalysis	Co@VMT MEM	10	99.70-100%	1.6 mg/cm ²	4.0-9.0	5 min	1.02	This work

REVIEWERS' COMMENTS

Reviewer #1 (Remarks to the Author):

The efforts of the authors involved in the manuscript are greatly appreciated. The manuscript has been carefully revised according to reviewers' comments. I recommend it for publication in Nature Communications.

Reviewer #3 (Remarks to the Author):

The authors exhaustively addressed all the comments.